# The impact of COVID-19 vaccination in the US: Averted burden of SARS-COV-2-related cases, hospitalizations and deaths

**Teresa K. Yamana**[1]*, **Marta Galanti**[1], **Sen Pei**[1], **Manuela Di Fusco**[2], **Frederick J. Angulo**[3], **Mary M. Moran**[3], **Farid Khan**[3], **David L. Swerdlow**[3], **Jeffrey Shaman**[1,4]*

**1** Department of Environmental Health Sciences, Mailman School of Public Health, Columbia University, New York, NY, United States of America, **2** Health Economics and Outcomes Research, New York, NY, United States of America, **3** Medical Development and Scientific/Clinical Affairs, Pfizer Vaccines, Collegeville, PA, United States of America, **4** Columbia Climate School, Columbia University, New York, NY, United States of America

* tky2104@cumc.columbia.edu (TKY); jls106@cumc.columbia.edu (JS)

**Data Availability Statement:** The data used in this analysis, as well as the model code, are publicly available at https://github.com/tkcy/avertedcases_code.

## Abstract

By August 1, 2022, the SARS-CoV-2 virus had caused over 90 million cases of COVID-19 and one million deaths in the United States. Since December 2020, SARS-CoV-2 vaccines have been a key component of US pandemic response; however, the impacts of vaccination are not easily quantified. Here, we use a dynamic county-scale metapopulation model to estimate the number of cases, hospitalizations, and deaths averted due to vaccination during the first six months of vaccine availability. We estimate that COVID-19 vaccination was associated with over 8 million fewer confirmed cases, over 120 thousand fewer deaths, and 700 thousand fewer hospitalizations during the first six months of the campaign.

## Introduction

By August 1, 2022, SARS-CoV-2, the virus responsible for the COVID-19 pandemic, had caused over 90 million cases and 1 million deaths in the United States [1]. While these numbers are likely affected by the widespread availability of SARS-CoV-2 vaccines, the precise impact of vaccination on the burden of COVID-19 disease is uncertain. Here we use a dynamic model, coupled with historical data, statistical inference methods, and hospitalization costs, to quantify the clinical and economic burdens of infections, hospitalizations, and deaths averted due to vaccination in the US, both cumulatively and in individual states, during the first approximately six months of vaccine availability when the wild type and alpha variants of SARS-CoV-2 were the predominant drivers of infection.

In mid-December 2020, the first SARS-CoV-2 vaccine received emergency use authorization in the US and was initially recommended for healthcare workers and long-term care facility residents, followed by adults aged 65 years and older, adults aged 16–64 with high-risk medical conditions and essential workers [2]. By early April 2021, the vaccine recommendation was extended to the general population aged 16 years and older. Subsequent steps have seen recommended vaccine use for 12–15 year-olds (May 2021) and 5–11 year-olds

**Funding:** This study was sponsored by Pfizer Inc.

**Competing interests:** TKY, MG, SP and JS are employees of Columbia University, which received funding from Pfizer in connection with the development of this study and of this manuscript. JS and Columbia University disclose partial ownership of SK Analytics. JS discloses consulting for BNI. MDF, FJA, MMM, and FK are employees of Pfizer and may hold stock or stock options. DS was employed at Pfizer at the time this work was conducted and he may own stock or stock options. This does not alter our adherence to PLOS ONE policies on sharing data and materials.

(November 2021). Three different vaccines (two mRNA vaccines and one antiviral vector vaccine) with varying efficacy and estimates of duration of protection have been authorized for use in the US. However, vaccination delivery has been variable: it was initially limited by vaccine availability, with roughly 15 million doses provided in the first month, but reached a peak of roughly 90 million doses administered during April, 2021 [1].

By early November 2021, 78% of the US population aged 12 years and older had received at least one dose of a SARS-CoV-2 vaccine, with heterogeneous distribution across age groups (97% of adults aged 65+ years vs 60% of persons aged 12–18 years) and across states (<65% in AL, ID, IN, LA, MS, ND, TN, WY, WV, compared with >90% in CT, MA, HI, VT, PA [2]). During the time period of vaccine rollout, variable levels of non-pharmaceutical interventions (NPIs), such as social distancing, closures of restaurants and bars, mask mandates and travel restrictions, were implemented across states with different start and end dates.

Here, we use a dynamic county-scale metapopulation model, previously used COVID-19 inference and projections [3–5], to conduct counterfactual simulations representing the effects of vaccination. These simulations are used to estimate the number of cases, hospitalizations, and deaths averted due to vaccination during the first six months of vaccine roll out.

## Methods

We used a metapopulation model with a Susceptible-Exposed-Infected-Recovered (SEIR) structure run at the county level, coupled with a data assimilation method (EAKF, the ensemble adjustment Kalman filter). We have previously used this framework for inference, forecasting and projections of influenza and SARS-CoV-2 infections at various locations and spatial scales, in both research and operational contexts [3–7]. Here, we simulated SARS-CoV-2 transmission within and among the 3142 counties of the United States.

We first used the model-inference system to fit reported case counts in each county of the US [3] from the time of identification of the first COVID-19 cases in the United States in February 2020, through December 14, 2020, the date of first authorized SARS-CoV-2 vaccination in the US. The inferred values of parameters and state variables on December 14, 2020 served as initial conditions for the averted burden analysis. The model estimates of susceptibility have previously been validated with serological data [4]. Specifically, estimates of cumulative infections during 2020, generated by this model when coupled with inference approaches, were compared and validated with estimates of seroprevalence derived from serological surveys collected on multiple dates and for multiple locations in the US. These external serological data provided strong validation of the model estimates of cumulative infections and thus population susceptibility.

We then added a representation of vaccination to the dynamical model structure using documented daily rates of vaccine administration [1, 8] (see S1 Text). State-level daily vaccination data from the CDC COVID Data Tracker [1] were allocated proportionally to each county based on population size. Within each county, we assumed equal probability of vaccination regardless of prior infection status. We modeled the vaccine as providing 90% effectiveness against infection [9–11]–i.e. 90% of vaccinated individuals with no prior immunity were fully protected from both infection and transmission, while the remaining 10% receive no protection from either infection or transmission. Specifically, 90% of vaccinated individuals with no prior immunity were removed from the Susceptible pool and placed in the Recovered compartment 24 days after administration of the first dose (S1 Text). Since effectively vaccinated individuals do not transmit the disease, we account for both direct and indirect effects of vaccination. In the Recovered compartment, we did not distinguish between vaccinated individuals and individuals recovered from infection; given uncertainty and limited data on re-infections

and waning, both were considered immune for the remainder of the simulation period. With the 24-day delay, the impact of vaccinations on the simulation begins on January 8th. This baseline scenario, retrospectively fitted to case counts, enabled estimation of the daily time-series of epidemiological parameters, including $R_t$, the time-varying reproductive number, for each county location from December 14, 2020 through June 3, 2021.

We ran the simulations through June 3, 2021 to focus on the impact of vaccination prior to the predominance of the Delta and Omicron variants [1]. Given that the higher transmissibility and immune escape properties of the Delta and Omicron variants require substantial additional modifications of the dynamical model structure, as well as re-parametrization, we restricted our analysis to the December 14, 2020 through June 3, 2021, or the pre-Delta, time period, during which the vaccine provided strong protection against infection. To quantify the burden averted by vaccination, we compared the baseline vaccination scenario to 3 counterfactual no-vaccination scenarios simulated over the same time period. All counterfactual scenarios assumed no vaccinations (or, equivalently, 0% vaccine effectiveness) but varied transmissibility to mimic different levels of non-pharmaceutical intervention (NPI) response in the absence of vaccination:

- Counterfactual Scenario 1; A no-transmission-change, no-vaccination scenario in which the $R_t$ daily time series for each location was as inferred for the baseline scenario;

- Counterfactual Scenario 2: A no-vaccination scenario in which $R_t$ for each location-day was increased 10% with respect to the baseline scenario; and

- Counterfactual Scenario 3: A no-vaccination scenario in which $R_t$ for each location-day was decreased 10% with respect to the baseline scenario.

These counterfactual scenarios represent potential population behaviors and policies that might have been effected in the absence of vaccination. Scenario 3 represents increased NPIs through policies and individual action; Scenario 2 represents a decrease of NPIs, perhaps due to pandemic fatigue. We compared cumulative SARS-COV-2 cases in the 3 no-vaccination scenarios to the baseline scenario at national and state levels, analyzed differences in averted cases among states, and identified factors correlating with vaccination success.

## Hospitalizations and deaths

To calculate hospitalizations and deaths in the counterfactual scenarios, we made the assumption that excess cases would have continued to lead to hospitalizations and deaths at the same overall rate as they did in each state during the summer and fall of 2020, prior to vaccine availability. We applied a state-specific pre-vaccine Case Hospitalization Rate (CHR) and a Case Fatality Rate (CFR) multiplier to the total number of averted cases in each scenario. These rates were computed by dividing the number of reported hospitalizations and deaths divided by the corresponding number of reported cases. The denominator for both the CHR and CFR in each state is the number of cases reported from August 1 –December 14, 2020 from the Johns Hopkins Center for Systems Science and Engineering (JHU CSSE) COVID-19 data set [12]. We assume a 4- and 9-day lag for hospitalizations and deaths, respectively, reflecting the delay from case reporting to hospitalization and death [13]. Note that the date of case reporting includes an additional lag from the time an individual first becomes infectious. Hospitalization data were compiled from the HHS dataset [14] and cases. August 2020 was the first full month with all states reporting daily COVID-19 hospitalizations. Death data are from the JHU CSSE COVID-19 Data [12]. We excluded deaths and cases prior to August 1, 2020, for consistency with the hospitalization data set, and because both the ascertainment rate (fraction of true

infections that are reported as confirmed cases) and the infection fatality rate (fraction of true infections that resulted in death) were unstable during the initial wave of the pandemic [4].

## Hospitalization costs

We calculated averted hospitalization costs by multiplying the distribution of estimated COVID-19 associated hospitalizations averted by the distribution of costs per hospitalization episode, obtained from the US-based Premier Healthcare COVID-19 claims database [15]. The median (interquartile range Q1-Q3) cost per hospitalization episode was $12,046 ($6,309-$25,361).

# Results

## Initialization

At the start of the simulation period, December 14, 2020, it was estimated 74.1% (95% credible interval: 70.2–78.6) of the US population was susceptible, 0.8% (95% CrI 0.6–1.2%) exposed, 0.8% (95% CrI 0.6–1.0%) infectious and 24.3% (95% CrI 19.2–28.6%) recovered. Fig 1 shows the distribution of the estimated epidemiological parameters across states at the beginning of vaccine administration. The median estimated susceptible fraction, corresponding to the fraction of the population that had not yet been infected since the beginning of the pandemic varied by state and ranged from 58% (95% CrI 56%-61%) in North Dakota to 94% (93%-95%) in Vermont. The susceptible fraction was highest in northwestern and northeastern states. The time dependent reproductive number ranged from median 0.8 (0.7–1.4 95% CrI) in Minnesota to 2.0 (1.7–2.3 95% CrI) in Tennessee. The CFR prior to the start of vaccination varied from 0.5% in Alaska (95% CrI 0.3–0.7%) to 2.3% (95% CrI 2.1–2.6%) in Rhode Island. The CHR in the same period ranged from 3.8% (95% CrI 3.6–4.0%) in Alaska to 20.7% in Kentucky (95% CrI 20.5–20.9%). While there was a modest reduction in CHR at the national scale from the pre-vaccine period (8.8% August 1, 2020 –December 14, 2020) to the analysis period (7.8% December 15, 2020 –June 2, 2021), we did not observe consistent population level differences in CFR at the national level (1.5% during both time periods) nor to CHR and CFR at the state level.

## Model results

Between December 14 and June 3, 2021, the baseline model estimated 16.1 million (95% CrI 15.1–18.3 million) total cases, 1.4 million (95% CrI 1.3–1.6 million) hospitalizations, and 246.7 thousand (95% CrI 230.4–279.6 thousand) deaths across the United States. These estimates were consistent with the 16.7 million cases, 1.3 million hospitalizations, and 250 thousand deaths reported in the JHU CSSE and HHS hospitalization datasets [12, 14]. S1 Fig shows fitted baseline cases compared to observations.

The time series of $R_t$ resulting from fitting the baseline scenario from December 14, 2020 through June 4, 2021 is shown at the state and national level in Fig 2. Note that $R_t$ in this analysis refers to the time-varying basic reproductive number, not to be confused with the effective reproductive number $R_{eff}(t)$, which is $R_t$ multiplied with the fractional susceptible population.

By June 4, 2021, 51% of the population in the US had received at least one dose of vaccine [1]. Vaccine coverage differed widely by location, ranging from 35% of the population in Mississippi up to 74% in Vermont. The weekly number of vaccinations administered increased over time: initially at less than 5 million vaccinated per week but reaching a peak of 14 million vaccinated per week in April when vaccination was extended to the general population aged 16 and older (S2 Fig).

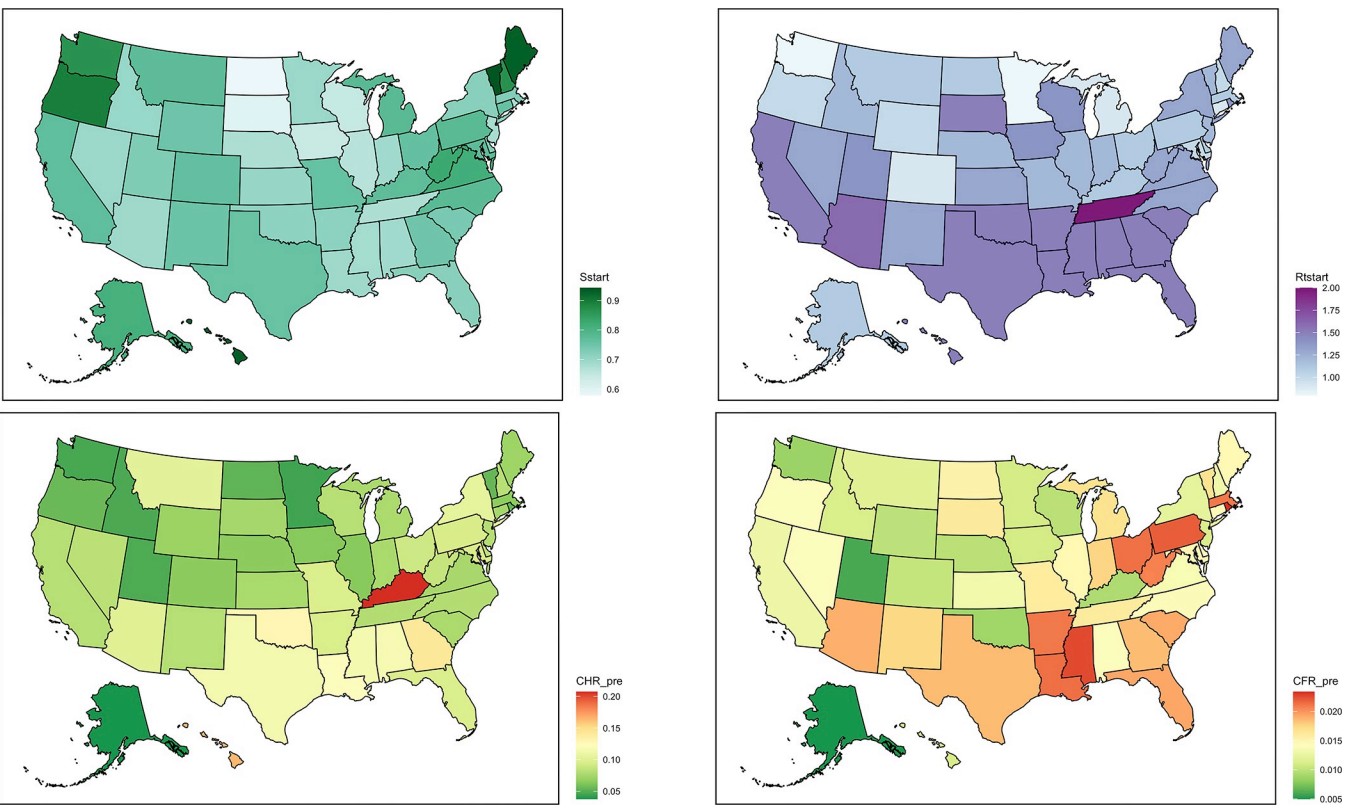

**Fig 1. Initial conditions, case hospitalization rates and case fatality rates.** Upper Left: Population susceptibility, S (proportion of the population not yet infected), at the start of vaccine administration; Upper Right: Time-varying reproductive number, $R_t$, at the start of vaccine administration; Lower Left: State-specific case hospitalization rate, CHR; Lower Right: State-specific case fatality rate, CFR. Color scales show the median values. Base maps show state boundaries from US Census Bureau [16].

Table 1 reports the cumulative averted COVID-19 cases, deaths, hospitalizations and hospitalization cost savings for the 3 scenarios, while Fig 3 shows the modeled COVID-19 case trajectories under the three counterfactual scenarios.

In the scenario with no change in transmission, we estimated that vaccination averted 8.1 million cases at the national level (median value, 95% CrI: [-4.8, 26.3] millions cases), 123.2 thousand deaths (median value, 95% CrI.: [-74.3, 403.0] thousand deaths) and 0.7 million hospitalizations (median value, 95% CrI: [-0.4, 2.3] millions hospitalizations). The median cost savings associated with averted hospitalization was $7.0 billion (median value, 95% CrI: [-$11.9, $112.0]) (see Table 1).

Increasing Rt by 10% with no vaccination in Counterfactual Scenario 2 roughly doubled the median cases averted nationally whereas decreasing Rt by 10% with no vaccination in Counterfactual Scenario 3 considerably reduced the averted burden during the approximately 6 months of analysis (Fig 3 and Table 1). In effect, the decreased Rt, representing increased NPIs, initially offsets the effects of no vaccination during the first 3 months when a more limited percentage of the population is effectively vaccinated. However, this effect decreases in mid-March as vaccination rates climb in the baseline scenario, and by May more cases are produced per day in Counterfactual Scenario 3 due to the absence of vaccination.

In all three counterfactual scenarios, the majority of averted cases occurred between April and June 2021 (Fig 3).

State-level results are presented in S1 Table and Fig 4. For individual states, the median estimates of cases averted under Scenario 1 ranged from roughly 1000 cases per hundred thousand

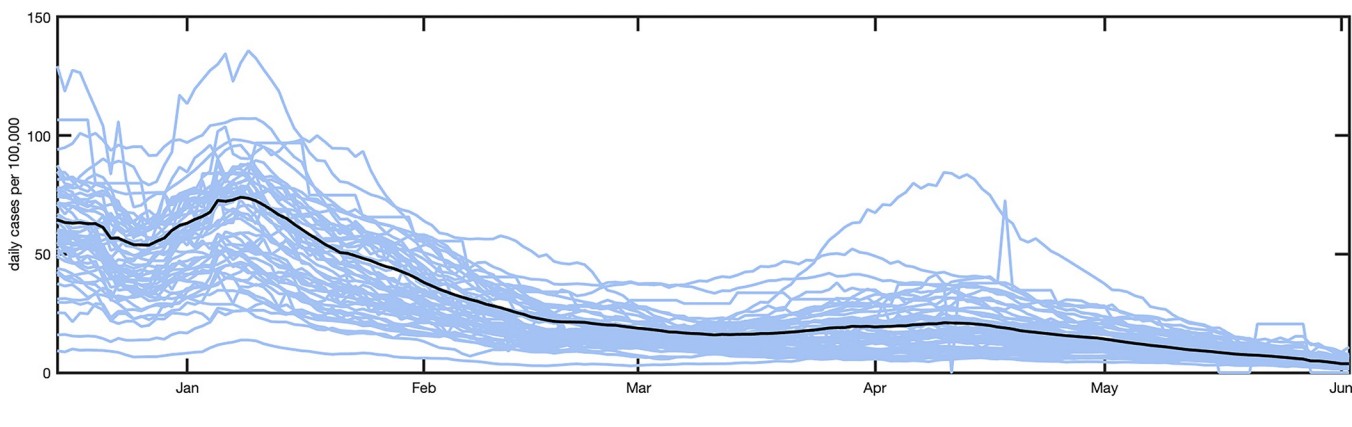

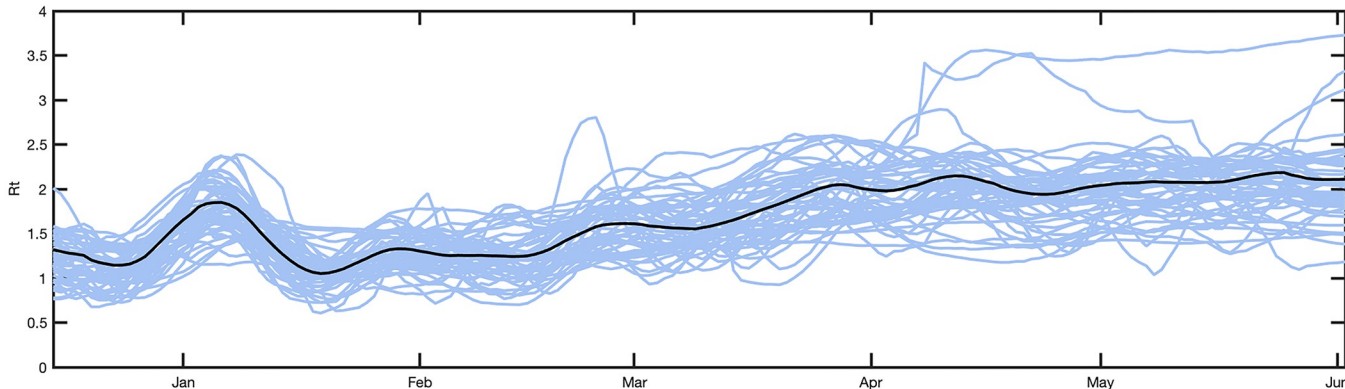

**Fig 2. Covid-19 cases and Rt from December 14th, 2020 through June 2nd, 2021.** Top: Covid-19 cases per 100,000 population per day (7-day moving average); Bottom: Median estimate of Rt in baseline scenario. Each blue line represents a single state. The black line is the overall national value. State and national estimates are derived by taking a population-weighted average of county-level estimates of Rt.

population in South Dakota to over 6000 cases per hundred thousand population in Maine and Arizona. Median cumulative averted hospitalizations varied from 74 per hundred thousand in South Dakota to 752 per hundred thousand in Kentucky. Median cumulative averted deaths varied from 16 per hundred thousand in South Dakota to over 120 per hundred thousand in Arizona and Rhode Island. Higher averted case burden correlated with higher vaccination rate ($R^2 = 0.16$) and higher population susceptibility at the beginning of the vaccination

**Table 1. Total cumulative COVID-19 cases, deaths, hospitalizations averted and hospitalization cost savings.**

|  | Scenario 1 | Scenario 2 | Scenario 3 |
|---|---|---|---|
|  | *No change in transmission* | *10% higher transmission* | *10% lower transmission* |
| Cases averted | 8.1 million | 17.0 million | -1.6 million |
| median, (95% CrI) | (-4.8, 26.3 million) | (1.5, 32.0 million) | (-8.8, 18.2 million) |
| Deaths averted | 123.2 thousand | 260.1 thousand | -25.1 thousand |
| median, (95% CrI) | (-74.3, 403.0 thousand) | (23.0, 489.7 thousand) | (-134.8, 278.8 thousand) |
| Hospitalizations averted | 0.7 million | 1.5 million | -0.1 million |
| median, (95% CrI) | (-0.4, 2.3 million) | (0.1, 2.8 million) | (-0.8, 1.6 million) |
| Hospitalization cost savings | $7.0 billion | $17.3 billion | -$0.9 billion |
| median, (95% CrI) | ($-11.9, 112.0 billion) | ($0.9, 170.3 billion) | (-$44.7, 70.1 billion) |

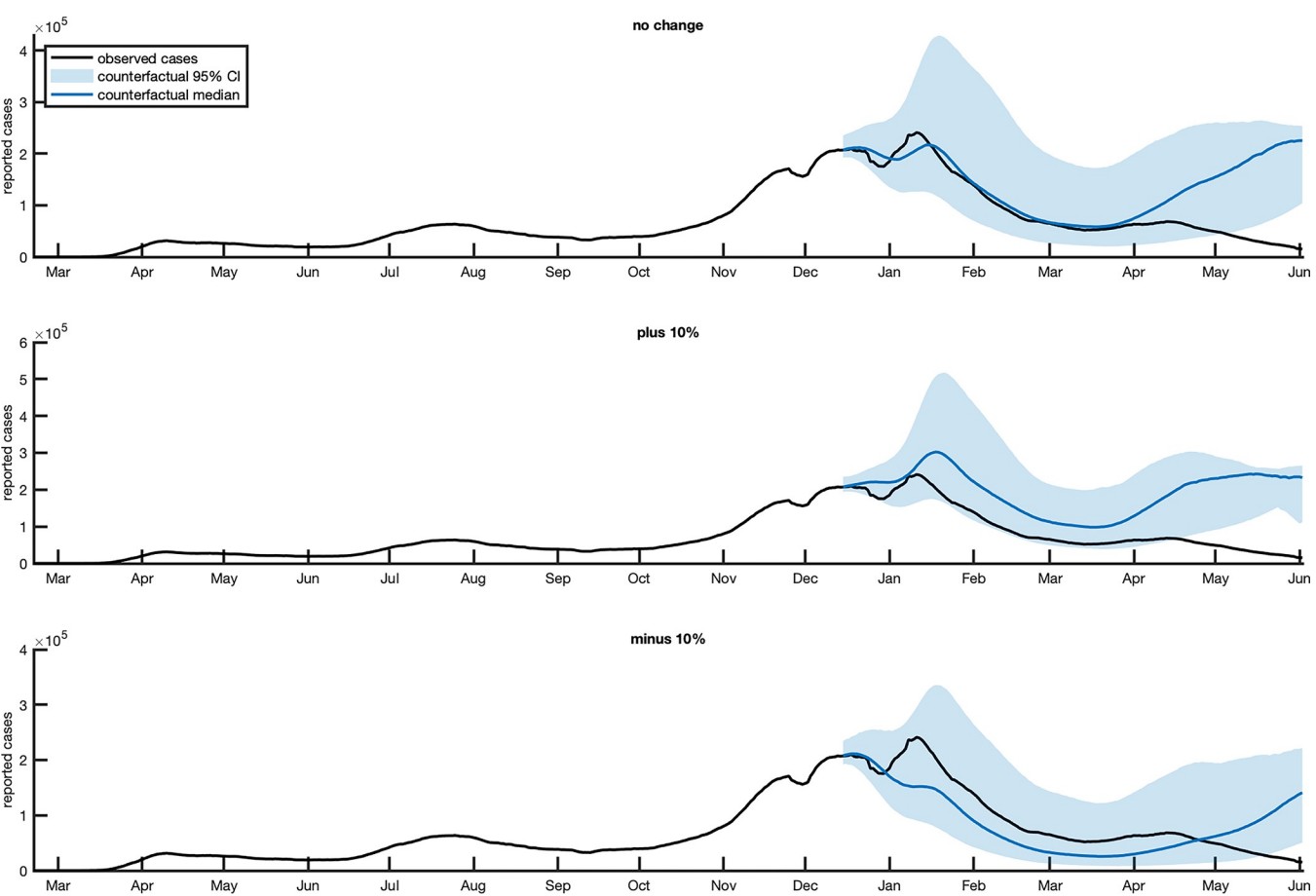

**Fig 3.** Modeled total COVID-19 Cases in Counterfactual Scenarios 1 (top panel), 2 (middle panel) and 3 (bottom panel) in the United States. The black line presents observed cases, the blue line indicates the median counterfactual projection, and the blue shaded area shows the 95% credible interval.

campaign ($R^2$ = 0.21). In most states, two to three times as many cases were averted under Scenario 2 compared to Scenario 1, indicating that the impact of vaccination would have been even greater if Covid restrictions were relaxed. Roughly three quarters of the states had fewer cases in Scenario 3 than in baseline, indicating that increasing non-pharmaceutical interventions would have been effective in averting cases during this time period.

## Discussion

Evaluating the population-level impact of COVID-19 vaccination through mathematical modeling can provide useful insights to policy makers. Here, we leveraged a validated dynamical modeling approach, previously used for research and operationally to simulate county-level COVID-19 transmission, to quantify the additional burden of disease in alternate scenarios without vaccination. Our analyses show that under unchanged NPI levels, COVID-19 vaccination in the US cumulatively prevented 8.3 million cases, 681 thousand hospitalizations and 118 thousand deaths in the first 6 months of implementation. States with high vaccination coverage such as Maine averted as many as 6,000 cases per 100,000 individuals.

These simulations are in general agreement with findings from three other modeling studies set in the US that have found substantial direct and indirect impacts of vaccination in terms of averted burden of disease. Shoukat et al. found that vaccination was fundamental for

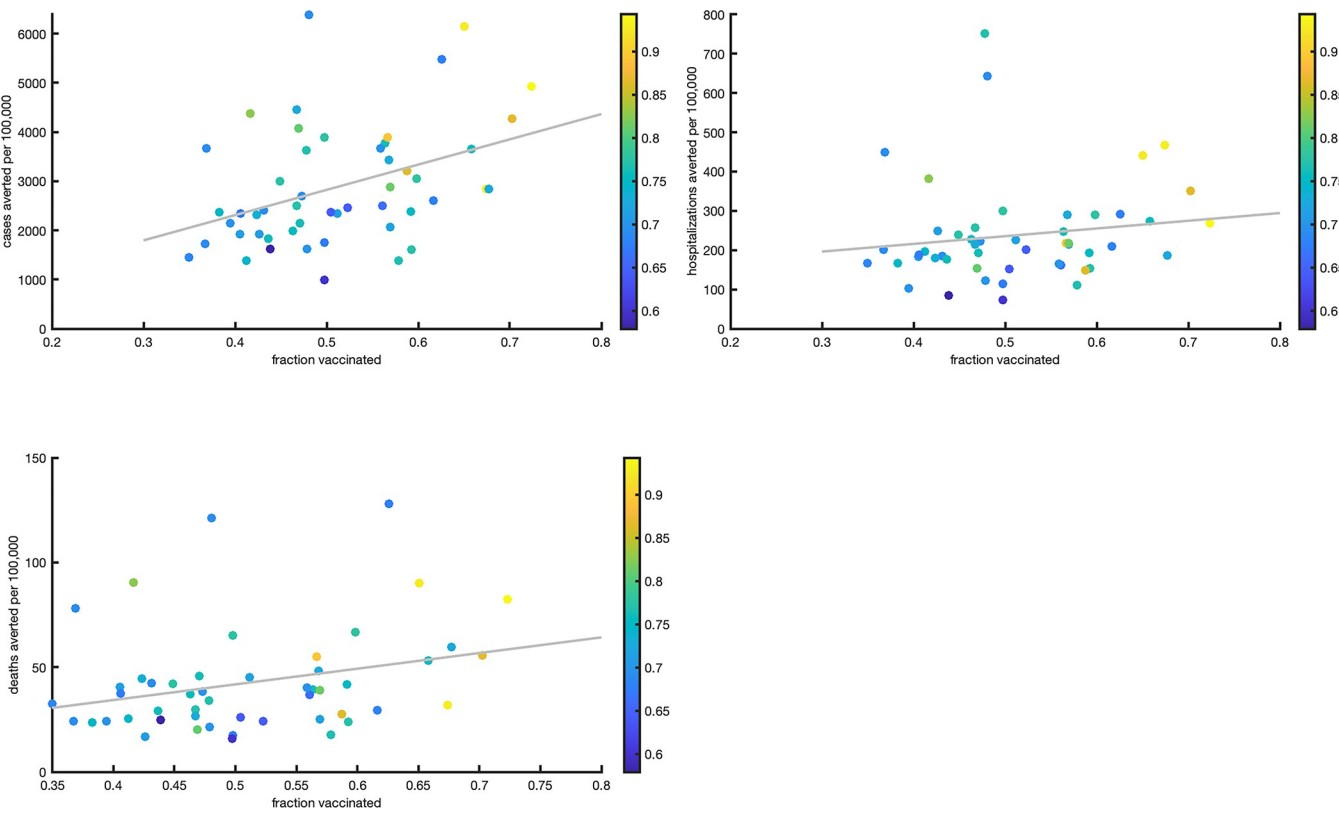

**Fig 4.** Total per capita averted cases (a), hospitalizations (b) and deaths (c) in each state between December 14, 2020 and June 2, 2021. The x-axis is the percent population vaccinated by June 2, 2021, and the y-axis is the averted cases/hospitalizations/deaths per 100,000 people. Each state is represented by a dot; the color scale of the dots indicate the estimated fraction of population susceptible at the beginning of vaccine rollout.

reducing the spring/summer wave in NYC by reducing cases by one third and hospitalization and deaths by half during December 2020 through July 2021 [17]. Vilches et al. showed that vaccination may have averted more than 14 million cases, 241 thousand deaths and 1.1 million hospitalizations in the US by late June 2021 [18]. Moghadas et al. found an even stronger effect of vaccination with 26 million cases, 1.2 million hospitalizations and 279,000 deaths averted through the end of June 2021 [19]. These three studies agree with our findings that indicate the US would have experienced a substantial wave of infections beginning in March/April 2021 in the absence of a vaccine [17–19]. Haas et al. [20] used a different methodology and found large direct effects of vaccination in Israel. This analysis compared the rates of SARS-CoV-2 infection-related outcomes between vaccinated and unvaccinated populations and estimated that two thirds of hospitalizations and deaths were averted with vaccination in the first four months of vaccine implementation [20].

Our study augments prior research in this field by providing further geographical granularity.

The state-level analyses provide a dynamic picture revealing trends and differences in the public health response to the COVID-19 pandemic, which may be informative for state and local policymakers. Additionally, our study sought to quantify the cost savings associated with vaccine-preventable severe disease (i.e. hospitalizations). The COVID-19 pandemic has challenged the capacity of hospitals and strained hospital and health system finances [21]. In Scenario 3, we estimated that, in the first six months of rollout of the COVID-19 Vaccination program, the hospitalization cost savings were $17.3 billion. More conservatively, Scenario 1

estimated cost savings of $7.0 billion. While these numbers are not adjusted for geographic differences, they provide a ballpark estimate that is in line with a previously published analysis, which estimated $13.8 billion of costs from preventable COVID-19 hospitalizations from June through November 2021 [22]. Our analyses show that the benefits of vaccination due to reduced hospitalizations translated into cost-savings in the billions of dollars. The broad availability of COVID-19 vaccines brought wide-ranging benefits from both a public health and economic perspective. Vaccination may also lessen other societal impacts associated with the pandemic (e.g. work productivity loss). The total economic impact may therefore be even greater than reported, and further studies elucidating those impacts are warranted.

Our study also adds to the existing literature by considering 3 counterfactual scenarios, all without vaccinations, but with varying $R_t$, that mimic different possible population responses to disease spread in the absence of a vaccine. The first counterfactual scenario is designed to quantify what the SARS-COV-2 related burden would have been without vaccinations if the population had maintained the same NPI measures as occurred with vaccination. The other two counterfactual scenarios are designed to explore the uncertainties of these estimates, as it is difficult to anticipate the public policy and population behavior response in the absence of a vaccine. Specifically, Counterfactual Scenario 2 represents a stronger relaxation of NPIs, possibly due to pandemic fatigue in the absence of an available vaccine, while Counterfactual Scenario 3 represents a reinforcement of NPIs during the 6 months of projections, assuming that the population would have responded with increased measures to control transmission. Counterfactual 3 shows that in the early months of the vaccine rollout, an increase in NPIs could have produced an even greater reduction of disease compared to vaccination as it occurred. However, while increased NPIs may have *slowed* transmission in the short term (the first months of vaccine rollout), those measures would not have been as effective as vaccination once the Alpha variant became established in the United States (Fig 3). The benefits of vaccination are seen in the difference between Counterfactual Scenario 3 and the baseline curve during the last month of simulation.

All 3 scenarios show that vaccination benefits were limited during the early months of vaccine rollout, and that most of the averted burden was realized in the last 2 months of the analyzed period. The winter peak of COVID-19 cases was reached in the US during mid-January 2021 just when the first vaccinations started to become effective. Vaccine availability constraints during the first months of the campaign restricted administration to portions of the population with increased risk of exposure and severe disease. It was not until April 2021 that vaccination was recommended for the general population aged $\geq$16 years. The combination of an initially slower rate of vaccination and a decreasing trend in transmission, with some states having a significant proportion of the population no longer susceptible to infection, narrowed the overall averted burden in the first months of 2021. Some exceptions occurred in states with larger initial susceptible fractions (e.g. Vermont); for these states the averted burden per hundred thousand was already significant in the early months of the vaccination campaign.

By March 2021, the Alpha variant, a SARS-CoV-2 strain with increased transmissibility relative to the wild type, became the predominant circulating serotype [1]. This variant, combined with progressive relaxation of NPIs in most states, likely produced the increase of $R_t$ inferred at this time. Simultaneously, the impact of vaccination, seen in the divergence between the baseline scenario and the no-vaccination scenario case curves, becomes much more evident at the national level (Fig 3).

A limitation of this analysis is that it relies on assumptions about whether and how the parameters inferred from the true observed course of the pandemic would have changed in the absence of a vaccine. Our primary counterfactual, Scenario 1, assumed that the parameters–including the disease transmission rates and the case ascertainment rate–would have been the

same with or without a vaccine. We explored some of the sensitivity to this assumption by altering the time-varying reproductive number in Counterfactual Scenarios 2 and 3. However, these are very simplified representations, and one could just as well imagine dramatically different counterfactual scenarios.

We limit our analysis to a relatively short projection time: the first six months of the vaccination campaign. Our estimates are therefore not generalizable to the entire period of the vaccination campaign. In subsequent months, booster doses, the expansion of the Pfizer vaccine to children aged 5–11, waning immunity, and the establishment of the more virulent Delta and immune-evading Omicron variants have made estimation of vaccine effects more challenging. These later phases of the pandemic driven by new variants led to tens of millions of Covid-19 infections. We are not able to say definitively whether the averted cases, hospitalizations and deaths quantified in this analysis were truly averted or merely delayed. Nevertheless, it can be argued that these early averted cases were crucial, as this period was prior to the widespread availability of antiviral medication and a time with substantially lower population immunity against severe outcomes.

Additional assumptions should also be noted. The model structure is parsimonious and does not explicitly represent certain factors including population age structure, breakthrough infections or reinfections. We used a constant case hospitalization rate (CHR) and case fatality rate (CFR) for each state, computed based on COVID-19 outcomes during the 6 months before vaccination, to calculate counterfactual hospitalization and deaths in all scenarios. These choices ignore differences in age-specific behavior and probability of severe outcomes. These assumptions may have led to biases in our results, as the risk of both hospitalization and death following COVID-19 infection are known to increase with age. Since our CHR and CFRs were calculated based on population-level averages, we likely underestimated the number of averted hospitalizations and deaths in the first few months of the analysis when vaccine uptake was concentrated in populations most at risk of severe outcomes. These assumptions also neglect spatial differences in CHR and CFR within a state.

We also note that the full effect of COVID-19 vaccination on hospitalizations and deaths derives from two effects: those averted due to averted cases; and those averted due to improved outcomes in vaccinated individuals if infected. The estimates of averted hospitalizations and deaths in this analysis are restricted to the effect of averted cases and do not include reductions in the probability of hospitalization and death among the vaccinated if infected–as a result, they likely underestimate the true number of averted hospitalizations and deaths. Each of the approved COVID-19 vaccines has been shown to be highly effective in preventing severe outcomes in individuals infected by SARS-CoV-2. Here, we assumed that the contribution of the second effect was relatively small compared to the first, as the vaccines were shown to be highly effective at preventing infections in the short term after inoculation and against the strains circulating at the time of the study [23–25].

In conclusion, our analysis shows that COVID-19 vaccination reduced the burden of disease. Base case results indicate that COVID-19 vaccination was associated with over 8 million fewer confirmed cases, over 120 thousand fewer deaths, and 700 thousand fewer hospitalizations in the first six months of the campaign. As such, COVID-19 vaccines represented a critical component of the public health response to the COVID-19 pandemic in the US.

## Supporting information

**S1 Text. Additional model details.**
(DOCX)

**S1 Fig. Time-series of confirmed Covid-19 cases in each state.** Observed data are shown in black (7-day backwards looking moving average) and fitted model results are in red.
(TIF)

**S2 Fig. Covid-19 cumulative vaccination rate from December 14th, 2020 through June 2nd, 2021.** The vaccination rate as a percentage of total state population is shown as indicated in the color bar.
(TIF)

**S1 Table. Estimated cumulative COVID-19 cases, deaths, hospitalizations averted by state.**
(XLSX)

## Author Contributions

**Conceptualization:** Teresa K. Yamana, Marta Galanti, Manuela Di Fusco, Jeffrey Shaman.

**Formal analysis:** Teresa K. Yamana, Jeffrey Shaman.

**Investigation:** Teresa K. Yamana.

**Methodology:** Teresa K. Yamana, Jeffrey Shaman.

**Project administration:** Jeffrey Shaman.

**Software:** Teresa K. Yamana, Sen Pei.

**Supervision:** Jeffrey Shaman.

**Validation:** Teresa K. Yamana.

**Visualization:** Teresa K. Yamana.

**Writing – original draft:** Teresa K. Yamana, Marta Galanti, Jeffrey Shaman.

**Writing – review & editing:** Teresa K. Yamana, Marta Galanti, Sen Pei, Manuela Di Fusco, Frederick J. Angulo, Mary M. Moran, Farid Khan, David L. Swerdlow, Jeffrey Shaman.

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
