## [Decision Letter · Decision Letter 0]

24 Oct 2022

PONE-D-22-26146The impact of COVID-19 vaccination in the US: averted burden of SARS-COV-2-related cases, hospitalizations and deathsPLOS ONE

Dear Dr. Yamana,

Thank you for submitting your manuscript to PLOS ONE. After careful consideration, we feel that it has merit but does not fully meet PLOS ONE’s publication criteria as it currently stands. Therefore, we invite you to submit a revised version of the manuscript that addresses the points raised during the review process.

Comments were kindly provided by two reviewers. I concur with Reviewer 1's suggestion that the authors clarify whether the indirect effects of vaccination were considered in the present study. If not, I would incline that authors can discuss any potential impact on the proposed result. Furthermore, please add a detailed method and validation of the proposed model for naive readers.

We look forward to receiving your revised manuscript.

Kind regards,

Sung-mok Jung

Academic Editor

PLOS ONE

Journal Requirements:

"TKY, MG, SP and JS are employees of Columbia University, which received funding from Pfizer in connection with the development of this study and of this manuscript.  JS and Columbia University disclose partial ownership of SK Analytics. JS discloses consulting for BNI. MDF, FJA, MMM, and FK are employees of Pfizer and may hold stock or stock options. DS was employed at Pfizer at the time this work was conducted and he may own stock or stock options."

"This study was sponsored by Pfizer Inc. TKY, MG, SP and JS are employees of Columbia

University, which received funding from Pfizer in connection with the development of this study

and of this manuscript. JS and Columbia University disclose partial ownership of SK Analytics.

JS discloses consulting for BNI. MDF, FJA, MMM, and FK are employees of Pfizer and may

hold stock or stock options. DS was employed at Pfizer at the time this work was conducted

and he may own stock or stock options."

"This study was sponsored by Pfizer Inc.  "

Reviewers' comments:

Reviewer's Responses to Questions

**Comments to the Author**

1. Is the manuscript technically sound, and do the data support the conclusions?

Reviewer #1: Partly

Reviewer #2: Partly

2. Has the statistical analysis been performed appropriately and rigorously? 

Reviewer #1: Yes

Reviewer #2: Yes

3. Have the authors made all data underlying the findings in their manuscript fully available?

Reviewer #1: Yes

Reviewer #2: No

4. Is the manuscript presented in an intelligible fashion and written in standard English?

Reviewer #1: Yes

Reviewer #2: Yes

5. Review Comments to the Author

Reviewer #1: Vaccination programs contributed to preventing the number of infected with SARS-CoV-2, hospitalizations and deaths related to COVID-19 worldwide. Quantifying the averted those burdens is crucial to evaluate the impact of the vaccination and decision-making at the time. Using the meta-population model, the authors tried to estimate the numbers of prevented SARS-CoV-2-related cases, hospitalizations and deaths attributable to vaccination in the US. Because the authors applied the model previously published in a few scientific journals, I believe the model could be robust even in the current context of the manuscript. However, I have some comments on the manuscript, so I would like the authors to consider them carefully.

In terms of the characteristics of the effectiveness of the vaccination, could the authors clarify it? The effectiveness would be divided into two: direct and indirect. I presume that the authors estimated the direct effect of vaccination because they did not take into account the reduction of the transmission rate of vaccinated people. I would be grateful if the authors discussed such effectiveness using the cited references (e.g., Israel’s study explored the direct effect but not the indirect one).

Please provide the fitted baseline scenarios for counties or states; otherwise, it is difficult to judge the scientific validation of the calibrated mode.

I would recommend that the authors provide information on the model even though the model was previously well described elsewhere. For example, how did the authors cooperate vaccination into the model? Unfortunately, I could not understand how they were considered even though I read the Supplementary Material. Please share the equation.

I agree with the advantages of the meta-population model that could capture the various characteristics of county or state-specific. However, some measures, e.g., CHR and CFR, are very important to consider the discrepancy between age groups. Therefore, please validate the usage of averages of such measures by showing the comparison between observed values and estimates of the hospitalizations and deaths from baseline scenarios, for example.

I do not understand why the authors calculated averted hospitalization costs. Please provide more insights obtained from the analyses in the discussion.

In the result, the authors showed that 74.1% were susceptible on 14 December 2020. It would be great if the authors compared this value and observed one, possibly from serological surveys, in order to validate the baseline model.

Please provide results shown in Table 1 for each county or state as tables or figures (cumulative) in the result or Supplementary Material.

Although the authors mentioned some results (i.e. averted cases, hospitalizations and deaths) estimated from previously published studies, there are no mentions of the methodology differences. For the direct effect of the vaccination, averted cases and hospitalizations can be calculated by the difference in incidences between unvaccinated and vaccinated. Why did the authors use the meta-population model? If the interactions between counties or states, why do not explore the insights (e.g. difference between states) further? Those points need to be described more in the manuscript.

Reviewer #2: This paper aims at caracterizing the averted covid-19 burden from vaccination in the first 6 months of the campaign in the US. It's a short paper, to the point, with a simple application of an already developed model. While the paper is suitable for publication in PLoS One, I think there are some barriers to have it published as it is.

Mainly, a lot of details are lacking in understanding the paper. Mainly the methods are not described fully (but available from other papers), and some diagnosis figures (posterior predictive checks or equivalent, vaccination baseline scenario model fit ...) are missing.

Some other remarks:

One question: why model at county scale but only present results at national or state-scale ?

« We modeled the vaccine as producing direct effects only » but with susceptible depletion, aren’t some indirect effects also taken into account in this model ?

A lot of simplifying assumptions are made but the discussion does not detail enough their possible impact on concusion (especially focusing only on the first six month without omicron, not using an age-stratified model, no discussion on delayed vs averted death, )

"All 3 scenarios show that vaccination benefits were limited during the early months of vaccine rollout" This might an artifact from the age-prioritization of covid-19 no ?

What makes the model finding so different from other modeling studies cited in the conclusion ?.

The paper is not standalone:

"and its [the model] full details are described in Pei & Shaman [4]."

"Parameter values for μ, Z, D and θ are assigned according to the values inferred in Pei and Shaman [4]."

Figure S1 is refered nowhere in the main text nor in the SI text.

So with the addition that the code is not shared yet, nor are figures of model fits, so it's really hard to provide a proper review of the paper. Especially as this study is sponsored by Pfizer, there should be a strong justification of the conclusions from the methods, which I cannot judge due to the lack of insight.

6. PLOS authors have the option to publish the peer review history of their article (what does this mean?). If published, this will include your full peer review and any attached files.

Reviewer #1: No

Reviewer #2: No

---

## [Author Response · Author response to Decision Letter 0]

30 Jan 2023

We thank the reviewers and the editor for their constructive feedback. We have addressed each of the comments in the manuscript, as well as in the responses below. Please note that this response is also attached to this submission as a word document with color-coded replies, for easier reading. 

Reviewer #1: Vaccination programs contributed to preventing the number of infected with SARS-CoV-2, hospitalizations and deaths related to COVID-19 worldwide. Quantifying the averted those burdens is crucial to evaluate the impact of the vaccination and decision-making at the time. Using the meta-population model, the authors tried to estimate the numbers of prevented SARS-CoV-2-related cases, hospitalizations and deaths attributable to vaccination in the US. Because the authors applied the model previously published in a few scientific journals, I believe the model could be robust even in the current context of the manuscript. However, I have some comments on the manuscript, so I would like the authors to consider them carefully.

1) In terms of the characteristics of the effectiveness of the vaccination, could the authors clarify it? The effectiveness would be divided into two: direct and indirect. I presume that the authors estimated the direct effect of vaccination because they did not take into account the reduction of the transmission rate of vaccinated people. I would be grateful if the authors discussed such effectiveness using the cited references (e.g., Israel’s study explored the direct effect but not the indirect one).

Thank you for raising this point – we have clarified this issue in the revised manuscript. We assumed vaccination to be fully protective against infection, so our analysis does include the indirect effects of vaccination. That is, because vaccinated individuals cannot be infected, they also cannot transmit the disease to others, thereby adding a level of protection to unvaccinated individuals. We have clarified this in the manuscript. We have also specified in the discussion which of the cited references consider direct vs indirect effects.

2) Please provide the fitted baseline scenarios for counties or states; otherwise, it is difficult to judge the scientific validation of the calibrated mode.

Thank you for this suggestion - the fitted baseline scenarios for each state have been added. 

3) I would recommend that the authors provide information on the model even though the model was previously well described elsewhere. For example, how did the authors cooperate vaccination into the model? Unfortunately, I could not understand how they were considered even though I read the Supplementary Material. Please share the equation.

We expanded the model description, which now includes the equation used to simulate vaccinate to the model description and is found in the Supplementary Material. We have also included a table of model parameters (Table S1). 

4) I agree with the advantages of the meta-population model that could capture the various characteristics of county or state-specific. However, some measures, e.g., CHR and CFR, are very important to consider the discrepancy between age groups. Therefore, please validate the usage of averages of such measures by showing the comparison between observed values and estimates of the hospitalizations and deaths from baseline scenarios, for example.

We have added a comparison of observed and estimated hospitalizations and deaths:

“Between December 14 and June 3, 2021, the baseline model estimated 16.1 million (95% CrI 15.1 – 18.3 million) total cases, 1.4 million (95% CrI 1.3 – 1.6 million) hospitalizations, and 246.7 thousand (95% CrI 230.4 – 279.6 thousand) deaths across the United States. These estimates were consistent with the 16.7 million cases, 1.3 million hospitalizations, and 250 thousand deaths reported in the JHU CSSE and HHS hospitalization datasets (1, 2).”

Additionally, we have added discussion on the impact of the age discrepancies. 

5) I do not understand why the authors calculated averted hospitalization costs. Please provide more insights obtained from the analyses in the discussion.

Thank you for your comment. We have further expanded the Discussion with the following: 

Additionally, our study sought to quantify the cost savings associated with vaccine-preventable severe disease (i.e. hospitalizations). The COVID-19 pandemic has challenged the capacity of hospitals and strained hospital and health system finances (3). In Scenario 3, we estimated that, in the first six months of rollout of the COVID-19 Vaccination program, the hospitalization cost savings were $17.3 billion. More conservatively, Scenario 1 estimated cost savings of $7.0 billion. While these numbers are not adjusted for geographic differences, they provide a ballpark estimate that is in line with a previously published analysis, which estimated $13.8 billion of costs from preventable COVID-19 hospitalizations from June through November 2021 (4). Our analyses show that the benefits of vaccination due to reduced hospitalizations translated into cost-savings in the billions of dollars. The broad availability of COVID-19 vaccines brought wide-ranging benefits from both a public health and economic perspective. Vaccination may also lessen other societal impacts associated with the pandemic (e.g. work productivity loss). The total economic impact may therefore be even greater than reported, and further studies elucidating those impacts are warranted.

6) In the result, the authors showed that 74.1% were susceptible on 14 December 2020. It would be great if the authors compared this value and observed one, possibly from serological surveys, in order to validate the baseline model.

The model estimates of susceptibility have previously been validated with serological data (5). Specifically, estimates of cumulative infections during 2020, generated by this model when coupled with inference approaches, were compared and validated with estimates of seroprevalence derived from serological surveys collected on multiple dates and for multiple locations in the US. These external serological data provided strong validation of the model estimates of cumulative infections and thus population susceptibility. 

7) Please provide results shown in Table 1 for each county or state as tables or figures (cumulative) in the result or Supplementary Material.

A table presenting the mean number of averted cases, hospitalizations and deaths in each location has been added to the Supplementary Material. 

8) Although the authors mentioned some results (i.e. averted cases, hospitalizations and deaths) estimated from previously published studies, there are no mentions of the methodology differences. For the direct effect of the vaccination, averted cases and hospitalizations can be calculated by the difference in incidences between unvaccinated and vaccinated. Why did the authors use the meta-population model? If the interactions between counties or states, why do not explore the insights (e.g. difference between states) further? Those points need to be described more in the manuscript.

We used the metapopulation model for this study as it has been extensively validated and used both for research and operational purposes over the course of the pandemic. In the revised manuscript, we have expanded our presentation of state-level results.

 

Reviewer #2: This paper aims at caracterizing the averted covid-19 burden from vaccination in the first 6 months of the campaign in the US. It's a short paper, to the point, with a simple application of an already developed model. While the paper is suitable for publication in PLoS One, I think there are some barriers to have it published as it is.

1) Mainly, a lot of details are lacking in understanding the paper. Mainly the methods are not described fully (but available from other papers), and some diagnosis figures (posterior predictive checks or equivalent, vaccination baseline scenario model fit ...) are missing.

Thank you for pointing out this oversight. We have added a figure showing baseline scenarios for each state. We have also added more information about the model structure and parameterization. 

Some other remarks:

2) One question: why model at county scale but only present results at national or state-scale ?

While our model simulates Covid transmission dynamics at the county scale, we use state level data to inform our CHR, CFR and vaccination rates. We chose to present results at state scale to avoid introducing further uncertainty into the model by disaggregating these state-level data. 

3) « We modeled the vaccine as producing direct effects only » but with susceptible depletion, aren’t some indirect effects also taken into account in this model ?

Thank you for raising this point – we should not have used the phrase ‘direct effects’ here, and have clarified this in the manuscript.

We assumed vaccination to be fully protective against infection, so our analysis does include the indirect effects of vaccination. That is, because vaccinated individuals cannot be infected, they also cannot transmit the disease to others, thereby adding a level of protection to unvaccinated individuals. 

4) A lot of simplifying assumptions are made but the discussion does not detail enough their possible impact on concusion (especially focusing only on the first six month without omicron, not using an age-stratified model, no discussion on delayed vs averted death, )

We have added the following discussion on delayed vs averted deaths:

“… the establishment of the more virulent Delta and immune-evading Omicron variants have made estimation of vaccine effects more challenging. These later phases of the pandemic driven by new variants led to tens of millions of Covid-19 infections. We are not able to say definitively whether averted cases, hospitalizations and deaths quantified in this analysis were truly averted or merely delayed. Nevertheless, it can be argued that these early averted cases were crucial, as this period was prior to the widespread availability of antiviral medication and a time with substantially lower population immunity against severe outcomes.”

We have also discussed the impact of non-age stratified model

“These assumptions may have led to biases in our results, as the risk of both hospitalization and death following COVID-19 infection are known to increase with age. Since our CHR and CFRs were calculated based on population-level averages, we likely underestimated the number of averted hospitalizations and deaths in the first few months of the analysis when vaccine uptake was concentrated in populations most at risk of severe outcomes.” 

5) "All 3 scenarios show that vaccination benefits were limited during the early months of vaccine rollout" This might an artifact from the age-prioritization of covid-19 no ?

We believe this finding is explained due to the relatively small proportion of vaccinated individuals, as well as the overall trend of declining transmission during that time.

6) What makes the model finding so different from other modeling studies cited in the conclusion ?.

Our findings are generally in agreement with previous studies cited in the manuscript. We have expanded our discussion of the differences in our methodological approach. 

Our study also adds to the existing literature by considering 3 counterfactual scenarios, all without vaccinations, but with varying Rt, that mimic different possible population responses to disease spread in the absence of a vaccine.

The paper is not standalone:

7) "and its [the model] full details are described in Pei & Shaman [4]."

8) "Parameter values for μ, Z, D and θ are assigned according to the values inferred in Pei and Shaman [4]."

We have removed the dependence Pei & Shaman by providing a full model description and a table of model parameters (Table S1). 

9) Figure S1 is refered nowhere in the main text nor in the SI text.

We added the reference in the methods.

10) So with the addition that the code is not shared yet, nor are figures of model fits, so it's really hard to provide a proper review of the paper. Especially as this study is sponsored by Pfizer, there should be a strong justification of the conclusions from the methods, which I cannot judge due to the lack of insight.

We hope that the revised materials provide confidence in the study methods and conclusions.

 

References

1. CSSE J. COVID-19 Data Repository by the Center for Systems Science and Engineering (CSSE) at Johns Hopkins University 2020 [Available from: https://github.com/CSSEGISandData/COVID-19.

2. Health UDo, Services H. COVID-19 reported patient impact and hospital capacity by facility 2020 [Available from: https://healthdata.gov/Hospital/COVID-19-Reported-Patient-Impact-and-Hospital-Capa/g62h-syeh.

3. Association AH. Massive growth in expenses and rising inflation fuel continued financial challenges for America’s hospitals and health systems. Cost of Caring Report Published April. 2022.

4. Amin K, Cox C. Unvaccinated COVID-19 hospitalizations cost billions of dollars. Health System Tracker Available at: https://bit ly/3GTceUq [cited 2021, Dec 10]. 2022.

5. Pei S, Yamana TK, Kandula S, Galanti M, Shaman J. Burden and characteristics of COVID-19 in the United States during 2020. Nature. 2021;598(7880):338-41.

---

## [Decision Letter · Decision Letter 1]

24 Feb 2023

PONE-D-22-26146R1The impact of COVID-19 vaccination in the US: averted burden of SARS-COV-2-related cases, hospitalizations and deathsPLOS ONE

Dear Dr. Yamana,

Thank you for submitting your manuscript to PLOS ONE. After careful consideration, we feel that it has merit but does not fully meet PLOS ONE’s publication criteria as it currently stands. Therefore, we invite you to submit a revised version of the manuscript that addresses the points raised during the review process.

We look forward to receiving your revised manuscript.

Kind regards,

Sung-mok Jung

Academic Editor

PLOS ONE

Journal Requirements:

Additional Editor Comments (if provided):

The manuscript has been well revised overall. However, I concur with Reviewer 1's comment regarding potential biases in CFR and CHR. Such naively calculated CFR and CHR may have been underestimated if the time delay from infection (or confirmation) to death (or hospitalization) was not fully taken into consideration, especially if the epidemic size exponentially increased in the corresponding period. Thus, I would like to strongly recommend that authors reestimate the CFR and CHR while taking the time delay distribution into consideration.

Reviewers' comments:

Reviewer's Responses to Questions

**Comments to the Author**

1. If the authors have adequately addressed your comments raised in a previous round of review and you feel that this manuscript is now acceptable for publication, you may indicate that here to bypass the “Comments to the Author” section, enter your conflict of interest statement in the “Confidential to Editor” section, and submit your "Accept" recommendation.

Reviewer #1: (No Response)

Reviewer #2: All comments have been addressed

2. Is the manuscript technically sound, and do the data support the conclusions?

Reviewer #1: Partly

Reviewer #2: Yes

3. Has the statistical analysis been performed appropriately and rigorously? 

Reviewer #1: Yes

Reviewer #2: Yes

4. Have the authors made all data underlying the findings in their manuscript fully available?

Reviewer #1: Yes

Reviewer #2: Yes

5. Is the manuscript presented in an intelligible fashion and written in standard English?

Reviewer #1: Yes

Reviewer #2: Yes

6. Review Comments to the Author

Reviewer #1: I am concerned about CHR and CFR. When the authors estimated CFR as dividing cumulative deaths reported between Aug 1 to Dec 14 2020 by cumulative cases infected during the same period, it may lead to an underestimate due to the right censoring issue of reported deaths (reporting delay between infection and death), especially in the case the magnitude of infections is large in the late period.

Reviewer #2: Thanks for including the code and for making the paper standalone. I appreciated the comments added in the description.

I do not have any comment that would require another review. Just two things, not mandatory

- l106: . "We modeled the vaccine as providing 90% effectiveness against infection"

I think "and transmission" should be added there to make sure there isn't any misunderstanding. (perhaps that could be also mentionned in discussion).

- fig S2: any tentatitive explanation of why Rt is higher in winter than in summer would be welcome. I do think this figures should be in the main text (with another pannel being vaccines uptake ?).

7. PLOS authors have the option to publish the peer review history of their article (what does this mean?). If published, this will include your full peer review and any attached files.

Reviewer #1: No

Reviewer #2: No

---

## [Author Response · Author response to Decision Letter 1]

4 Apr 2023

We thank the editor and reviewers for your time and comments. We have addressed each of the comments below, and in the manuscript.

Editor:

The manuscript has been well revised overall. However, I concur with Reviewer 1's comment regarding potential biases in CFR and CHR. Such naively calculated CFR and CHR may have been underestimated if the time delay from infection (or confirmation) to death (or hospitalization) was not fully taken into consideration, especially if the epidemic size exponentially increased in the corresponding period. Thus, I would like to strongly recommend that authors reestimate the CFR and CHR while taking the time delay distribution into consideration.

Thank you for raising this issue. We fully agree that the delays between case reports and the corresponding hospitalizations and deaths are important. We did in fact apply delays, but this was not clearly explained in our original manuscript. We have edited the manuscript to be more precise. 

Reviewer #1: I am concerned about CHR and CFR. When the authors estimated CFR as dividing cumulative deaths reported between Aug 1 to Dec 14 2020 by cumulative cases infected during the same period, it may lead to an underestimate due to the right censoring issue of reported deaths (reporting delay between infection and death), especially in the case the magnitude of infections is large in the late period.

Thank you for raising this issue. We fully agree that the delays between case reports and the corresponding hospitalizations and deaths are important. We did in fact apply delays, but this was not clearly explained in our original manuscript. We have edited the manuscript to be more precise. 

Reviewer #2: Thanks for including the code and for making the paper standalone. I appreciated the comments added in the description.

I do not have any comment that would require another review. Just two things, not mandatory

- l106: . "We modeled the vaccine as providing 90% effectiveness against infection"

I think "and transmission" should be added there to make sure there isn't any misunderstanding. (perhaps that could be also mentionned in discussion).

Thank you for your comments. We have added ‘and transmission’ to lines 107 and 108 to avoid any misinterpretation. 

- fig S2: any tentatitive explanation of why Rt is higher in winter than in summer would be welcome. I do think this figures should be in the main text (with another pannel being vaccines uptake ?).

We moved the figure to the main text. We provide some possible explanations for temporal changes in Rt in the discussion (eg population behavior, new variant, Lines 387-390). While we’re very interested in other possible factors, this does not fall within the scope of this study.

---

## [Editor Report · Decision Letter 2]

12 Apr 2023

The impact of COVID-19 vaccination in the US: averted burden of SARS-COV-2-related cases, hospitalizations and deaths

PONE-D-22-26146R2

Dear Dr. Yamana,

We’re pleased to inform you that your manuscript has been judged scientifically suitable for publication and will be formally accepted for publication once it meets all outstanding technical requirements.

Kind regards,

Sung-mok Jung

Academic Editor

PLOS ONE

Additional Editor Comments (optional):

I appreciate the authors’ effort in integrating all the comments in the manuscript. The manuscript has been well-revised, and in my opinion, it is now ready for acceptance. However, if the authors could clarify how exactly the reporting delay was considered in the calculation of CFR and CHR (e.g., assuming identical delays across all cases or back-projecting the epidemic curve with the distribution), that would be more helpful for naïve readers to follow. Again, congratulations!
---

## [Editor Report · Acceptance letter]

17 Apr 2023

PONE-D-22-26146R2 

The impact of COVID-19 vaccination in the US: averted burden of SARS-COV-2-related cases, hospitalizations and deaths 

Dear Dr. Yamana:

I'm pleased to inform you that your manuscript has been deemed suitable for publication in PLOS ONE. Congratulations! Your manuscript is now with our production department. 

Kind regards, 

on behalf of

Dr. Sung-mok Jung 

Academic Editor

PLOS ONE